# Reliability and Validity of the Six Spot Step Test in People with Intellectual Disability

**DOI:** 10.3390/brainsci11020201

**Published:** 2021-02-06

**Authors:** María Mercedes Reguera-García, Raquel Leirós-Rodríguez, Eva Fernández-Baro, Lorena Álvarez-Barrio

**Affiliations:** 1SALBIS Research Group, Department of Nursing and Physiotherapy, Faculty of Health Sciences, Universidad de León, 24400 Ponferrada, Spain; mercedes.reguera@unileon.es; 2Asprona Bierzo, Ave 3rd 24, 24400 Ponferrada, Spain; evafb@aspronabierzo.org; 3Department of Nursing and Physiotherapy, Universidad de León, 24400 Ponferrada, Spain; lalvb@unileon.es

**Keywords:** clinical test, walking ability, postural control, motor control, balance, evaluation

## Abstract

Clinical tests for the evaluation of balance in people with intellectual disability that have been most commonly used depend on the subjective evaluation of the evaluator, easily reach the ceiling effect and are poorly sensitive to small changes; but new tests have been developed, such as the Six Spot Step Test. The aim of this study was to determine the validity and within-day and day-to-day test–retest reliability of the Six Spot Step Test in people with intellectual disability. A descriptive cross-sectional study was conducted with 18 people with intellectual disability. The participants conducted the Six Spot Step Test three times and a set of five clinical tests for the balance assessment. The relative reliability was excellent (Intraclass Correlation Coefficient (ICC) = 0.86 − 0.97), and the absolute reliability ranged between 4.7% and 7.3% for coefficient variation and between 0.6 and 1.2 for the standard error of measurement. Linear regression models showed that that test can explain the results of the Timed Up & Go, Four Square Step Test and the Berg Balance Scale. The Six Spot Step Test proved to be as valid and reliable for the evaluation of dynamic balance in people with intellectual disability as the most frequently used tests for the clinical evaluation of postural control.

## 1. Introduction

People with intellectual disability (ID) frequently show, to a greater or lesser extent, delays in the maturation and development of motor control [1,2,3]. Among the motor skills that govern motor control, postural control is one in which people with ID present more limitations [4]. This is due to the fact that the physiopathology of ID usually involves a certain degree of incomplete development of the central nervous system, which controls motor and cognitive functions [5,6]. Moreover, these people show a premature ageing process [6] and, as in the case of healthy people, their static and dynamic balance worsen with age, due to the deterioration of the different subsystems of postural control (mainly the somatoaesthetic, vestibular and visual subsystems) [7,8].

Consequently, the risk of falling in these people is greater [9] and, thus, they especially benefit from the early detection of balance deterioration [4]. Previous studies have suggested that gait deceleration can be an early indicator of balance deterioration [10,11]. Therefore, the evaluation of these conditions is relevant in people with ID. The clinical tests for the evaluation of balance and gait that have been most commonly used in this population are: the Berg Balance Scale (BBS) [12,13,14], the Tinetti Scale [4], Single-leg Stance [13,14], the Functional (FRT) [13,14] and Lateral Reach Test (LRT) [13] and Timed Up & Go (TUG) [13,14]. All these tests have the advantage that they are low cost. However, in contrast, they depend on the subjective evaluation of the evaluator, easily reach the ceiling effect and are poorly sensitive to small changes in the postural control system [15]. On the other hand, new tests have been developed in the field of neurology that require the completion of more complex tasks, such as the Six Spot Step Test (SSST) [16]. This test involves fast criss-cross walking along a 5 m rectangular course, while kicking five blocks out of circles marked on the floor. Although its use has increased in the last years, SSST is not yet widely adopted in research and clinical practice [17]. One of the advantages of this test is that it is more demanding in the challenge of the components of coordination, dynamic balance, lower limb strength, eye-motor coordination and cognition than the previously mentioned tests [16,18]. That is, the SSST requires the subject to walk while hitting the blocks, demanding the performance of two tasks simultaneously (a condition that affects balance control) [19]. However, the psychometric properties of SSST, although promising, have not been completely established [17,20].

Furthermore, since SSST involves measuring the time needed to complete the evaluated task, it has been questioned whether conducting its evaluation at the same time as the task is performed was reliable enough, and whether it would be more reliable to evaluate a video recording of the task, since the participant would not be required to repeat the test. That is, the evaluation of the SSST implies that the evaluator is attentive to the moment in which the patient hits the last block to determine the time by chronometer simultaneously. This high demand of the evaluator by the SSST can increase its reliability if it is quantified later through the analysis of the video recording of the test.

For all of the above, the aims of this study were: (a) to determine the within-day and day-to-day test–retest reliability of SSST in people with ID; (b) to determine the reliability of the test if it is quantified with a chronometer simultaneously and later through video; and (c) to evaluate its validity. This work was based on the hypotheses that: (a) it is a reliable test for the evaluation of dynamic postural control in people with ID; (b) that its direct quantification using a chronometer is reliable; and (c) that it is a valid test for the quantification of postural control.

## 2. Materials and Methods

### 2.1. Study Design and Sample

A transversal, descriptive study was conducted with participants selected by convenience sampling. All participants belonged to a protective association for people with ID. They lived in flats supervised by such an association and performed different socioeducational tasks. The inclusion criteria were as follows: (a) capacity to stand and walk 100 m autonomously; (b) enough intellectual capacity to follow the verbal instructions involved in the evaluation procedure; (c) presence of moderate or mild intellectual and physical disability operationalised by scoring 60–90 points in the General Functionality item of the Inventory for Client and Agency Planning (ICAP); and (d) a score above 24 points in the Mini-Mental State Examination (MMSE) [21]. The exclusion criteria were: (a) the need to use orthopedic devices for walking; (b) current diagnosis of traumatic pathology; and (c) diagnosis of visual, auditory or vestibular alterations that affect balance. In Spain, the National Institute of Statistics indicates that the total population of people with ID is 3848 people [22]. The sample selection process concluded with the inclusion of 18 participants (7 women: 38.9%), which implies that the results of this research have a margin of error of 17% and a confidence level of 84%.

The participants and their legal guardians were previously informed about the objectives of this study and the evaluation procedure that they would be subjected to. If they agreed to participate, in accordance with the Declaration of Helsinki (rev. 2013), all participants (or their legal guardians) signed an informed consent prior to their participation in the study. The institutional review board approved the study protocol and granted the ethical approval from the Ethics Committee of the University of León (ETICA-ULE-030-2019).

### 2.2. The SSST

The SSST consisted in timing the completion of four races. Each race was performed by criss-cross walking a rectangular track and pushing out wooden blocks situated in circles throughout the track [23]. Firstly, the blocks were pushed out twice with the right leg and, then, the blocks were pushed out twice with the left leg. The blocks had to be pushed out with slight kicks using the lateral or medial part of the foot. With the right foot, two blocks by the right side were pushed out with the lateral part of the foot, and two blocks by the left side were pushed out with the medial part of the foot. Subsequently, the test was repeated with the left foot (Figure 1).

The participants were asked to conduct it as fast as they could. Before recording the performance of the tests, a familiarisation test was carried out to verify that the participant understood the test. The test protocol was followed in compliance with the recommendations of Callesen et al. [17]. If the participant used the wrong leg or the tip of the foot to push out one or more blocks, the test was repeated until it was performed correctly.

### 2.3. Procedure

The evaluators were physiotherapists with extensive experience in evaluation, and they received two 3 h training sessions about the procedure. Each researcher was in charge of the same test throughout the entire study.

An initial interview was held to gather information about the participants’ medical-surgical history and personal data, as well as to apply the MMSE and the ICAP. Then, the anthropometric measurements were recorded (weight and height) using a Seca height rod and scale (SECA^®®^, Hamburg, Germany), which were used to calculate the body mass index (BMI).

The participants conducted the SSST three times: two times in one day, with a 5 min separation (within-day reliability) and then a third time 48 h after (day-to-day and inter-evaluator reliability). The three attempts of the test were recorded in video using an iPhone SE (Apple Inc, California, CA, USA) and were analysed by a different evaluator, who was blinded to the results recorded using a chronometer (inter-evaluator reliability). To record the time, the evaluator who explained the test used the chronometer of an iPhone 7 plus (Apple Inc., California, CA, USA).

In the intermediate day between the days in which the SSST was conducted, the participants carried out a set of clinical tests for the evaluation of postural control. The sequence of application of the tests was as follows:

(a) TUG: this recorded the time (in seconds) that the participants took to rise from an armchair, walk 3 m, navigate an obstacle on the floor and return to a fully seated position in the chair [24]. This test has shown excellent reliability (Intraclass Correlation Coefficient, ICC = 0.98) in healthy adults and in individuals with cerebral palsy, multiple sclerosis, Huntington’s disease, post-stroke, and spinal cord injury [25].

(b) Four Square Step Test (FSST): this measures the dynamic balance and postural orientation by quantifying the seconds required to conduct a motor and cognitive task [26]. The motor task consists in walking across a set of sticks placed on the floor (making four separated squares). The participant begins in one square and must advance toward each of the four squares, and then reverses direction to go back to the starting point [27,28]. This test has shown good–excellent reliability in adult populations with different pathologies (ICC = 0.73–0.98) [28,29].

(c) FRT: this test evaluates the postural control of the trunk in the antero–posterior axis and has been identified as a useful test to detect balance deterioration in people with disabilities. The participant starts from a stable sitting position with his/her arm extended anteriorly, 90° shoulder flexion and hand closed. The score is the difference between the length of the arm and the maximum distance reached, using a fixed line on the wall [30]. This test has obtained excellent reliability values (ICC = 0.9–0.98) [31,32].

(d) LRT: this test evaluates the postural control of the trunk in the mid-lateral axis. It is performed by arranging the sitting person with his/her dominant arm extended, 90° shoulder abduction, hand closed and his/her contralateral arm resting on his/her body. The participants received the following standardised instructions: they had to move the arm laterally as far as possible, without losing their sitting balance. Their hips had to remain in full contact with the surface of the pressure mapping device and no trunk flexion or rotation was allowed. The maximum perceived position was maintained for three seconds before returning to the starting position. The hand excursion was recorded laterally from the tip of the third finger. A high correlation between the LRT result and the center of body pressure was obtained [33].

(e) BBS: this test evaluates the functional limitations associated with activities of daily living that require balance, such as reaching, flexing, transferring and standing, among others. Such tasks are distributed in 14 items, which are scored between 0 and 4 points, with a total score of 56 points [34]. This test has shown excellent reliability in elderly people (ICC = 0.95–0.98) [35,36].

### 2.4. Statistical Analysis

A descriptive analysis of all the study variables was performed through the calculation of the mean values, standard deviations and the 95% confidence interval (CI).

The reliability analysis between parallel tests was conducted through the correlation analysis between the results obtained with the chronometer (live observation) and the observation of the video recordings (later observation). The relative reliability was analysed using ICC and their 95% CI (values less than 0.5, between 0.5 and 0.75, between 0.75 and 0.9, and greater than 0.9 are indicative of poor, moderate, good and excellent reliability, respectively) [37].

To measure the absolute reliability, we used the coefficient variation (CV) [38] and the standard error of measurement (SEM and %SEM) [39]. Agreement between repeated measures was analysed using the Bland–Altman method, which provides insight into the agreement by calculating the mean difference between two sets of observations based on the mean values. The range of 1.96 × SD above and below the mean difference was defined as the 95% limits of agreement (LOA). In addition, LOA is also presented as a symmetric variation on an assumed mean difference of zero. The within-day agreement of the SSST was determined from the two tests that were performed on the first day. The day-to-day agreement of the SSST was determined from the first test on Day 1 and the last test on Day 3.

To evaluate the validity of the SSST test, its result was correlated with that of the other clinical tests of postural control evaluation used in this study. Moreover, we applied linear regression models using the SSST result (dependent variable) and clinical tests of postural control evaluation (independent variables), adjusted by age. To evaluate the fit in the linear regression models, the R^2^ statistic was used. The criteria to evaluate the adjustment values higher than 0.25 were used when they were significant. All calculations were performed using the STATA software v.13 (Stata Corp., College Station, TX, USA). The significance level was set at *p* < 0.05.

## 3. Results

### 3.1. Descriptive Analysis

The sample had an average age of 31.8 ± 9.9 years and showed BMI values of normal weight for the subgroup of men and overweight for the subgroup of women (Table 1). The comparisons between the two sexes were statistically significant for weight and BMI (*p* < 0.05).

### 3.2. Test–Retest within-Day and Day-to-Day Reliability Analyses

Table 2 shows the results obtained through the two measurement methods applied. No statistically significant results were obtained in any case. However, it can be observed that, although without statistically significant differences, there were higher scores in the third attempt of the test (especially in the results obtained using the chronometer).

Table 3 shows that the relative reliability was excellent in all the studied conditions (ICC = 0.86–0.97), with a higher within-day reliability in both measurement methods. The absolute reliability tests ranged between 4.7% and 7.3% for CV and between 0.6 and 1.2 for SEM. In both cases, the lowest variability value corresponded to the within-day analysis of the evaluation with the chronograph, whereas the highest value corresponded to the day-to-day analysis of the same evaluation method. The differences of means between the tests ranged from −0.26 to −0.99, with the highest variability being obtained in the within-day evaluation of the video measurement. Figure 2 shows the variation of the differences of means and the average score of means of the within-day reliability (Figure 2A) and day-to-day reliability (Figure 2B) obtained using the chronometer.

The reliability analysis through parallel tests showed high similarity between the results obtained with the chronometer and video observations (Table 4).

### 3.3. Validity Analysis

The result of the SSST obtained in the first attempt quantified using the chronometer was significantly correlated with TUG (r = 0.56; *p* < 0.04), FSST (r = 0.93; *p* < 0.001), FRT (r = −0.52; *p* < 0.03) and BBS (r = −0.67; *p* = 0.008), but not with LRT (*p* > 0.05). The result of the SSST obtained in the first attempt quantified through the observation of the video recordings showed exactly the same correlation results.

The linear regression models showed that the SSST can explain the results of the other clinical tests applied in this study (Table 5). It was observed that the tests whose results were explained by the SSST to a greater extent were: FST, TUG and BBS (*p* < 0.05). In every case, the results were better for the models generated with the results of the SSST obtained through video observation.

## 4. Discussion

The aim of this study was to determine the test–retest reliability of the SSST in people with ID and evaluate its validity. The obtained results indicate that the reliability of the test, both within-day and day-to-day, is excellent, regardless of whether the task is evaluated in situ or through video recording.

In this study, the results of the SSST obtained lower values compared to other studies conducted in populations with neurological pathology [16,17] and higher values with respect to healthy people [40]. High consistency was obtained between the repeated tests, and the reliability was slightly lower compared to previous studies [16,18], with the within-day reliability being higher than the day-to-day reliability with both the chronograph and the video recording. This is in line with the findings of a different study in people after stroke [19], although with reliability values higher than those identified in people with Parkinson’s disease [41].

The results obtained with the chronograph and video recording revealed that the CV values and mean difference were similar in all conditions, except for within-day video recording. This phenomenon is probably due to the fact that video recording can obtain more accurate scores, since it captures the exact moments of the start and end of the task. However, the results obtained in the SEM% below 10% in all the evaluation conditions indicate that the clinical sensitivity was excellent [42]. Therefore, the need for additional material (video camera) and time (for the subsequent visualisation of the video) does not justify the use of this evaluation method.

The SSST has shown reliability properties equivalent to those of TUG in samples with similar characteristics [13,43]. However, it must be taken into account that the requirements to complete the SSST are more demanding than those for TUG, since the former requires maintaining single-leg stance and eye-motor coordination with the foot. As in the case of FRT [32,44] and LRT, the SSST showed equivalent psychometric properties, although it is important to take into account that the information of these other two tests is more limited by the evaluation of a much simpler task in terms of understanding and execution.

Regarding the FSST, the reliability results found in the literature are diverse, since they have been evaluated in samples with very diverse characteristics (older adults, Parkinson’s disease, Huntington’s disease, multiple sclerosis patients, etc.) [30]. In any case, the SSST has also shown psychometric properties equivalent to those of the FSST, although it must be taken into account that the requirements of the patient’s understanding and attention to be evaluated with the FSST are more demanding. In comparison with the BBS [35,36], the SSST also reached similar reliability values. However, in contrast to the previous cases, the physical and intellectual demands for the successful execution of the SSST are lower. The SSST is completed in less time and does not present a ceiling effect in its evaluation; however, on the other hand, it provides less detailed information about the postural control subsystems that may be causing the loss of balance.

In any case, the strong correlation between the SSST and the rest of the clinical tests for the evaluation of postural control used in this study indicates that the construct assessed by this test is also dynamic balance. Moreover, these results were corroborated with the generation of the linear regression models.

Therefore, considering that the clinical and non-instrumental evaluation of balance still lacks a gold standard [45,46] with which to compare the SSST, it could be asserted that the SSST is as valid for the evaluation of balance in people with ID as other tests, such as BBS and TUG. Although, the SSST presents a higher ceiling effect and, therefore, it is more sensitive to small signs of balance deterioration. Consequently, the SSST combines the advantage of the short completion time presented by TUG with the demand of more complex tasks than BBS, thus enabling the early detection of balance alterations.

At the same time, the application of the SSST could be complemented with the FSST, since the latter evaluates the influence of the simultaneous execution of a cognitive task on postural control. Lastly, simpler tests with lower psychometric properties, such as FRT and LRT, should be taken into account only when the evaluation of the postural control of the trunk is a priority or when the degree of ID of the patient does not allow for conducting other tests or scales that involve greater understanding and collaboration.

This study shows methodological limitations that must be pointed out. The main limitation was the small sample size. Future research, in addition to evaluating larger samples and patients with other neurological pathologies, should be focused on longitudinal studies that correlate the result obtained in the SSST and the falls suffered by the patients. Similarly, it is important to highlight the novelty of this topic and the great possibilities for future research in this field to contribute to the development of low-cost clinical evaluation tools of short application time, which would directly improve the quality of life of people with ID.

## 5. Conclusions

The SSST showed excellent within-day reliability and good day-to-day reliability. This test proved to be as valid and reliable for the evaluation of dynamic balance in people with ID as the most frequently used tests for the clinical evaluation of postural control, such as BBS and TUG. Furthermore, the SSST has the advantage of presenting a higher ceiling effect and requiring less time to complete than TUG, as well as demanding more complex tasks than BBS, which will allow for an earlier detection of balance alterations.

## Figures and Tables

**Figure 1 brainsci-11-00201-f001:**
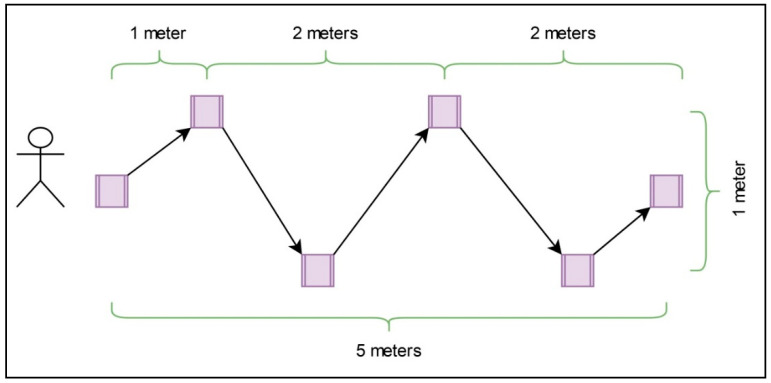
Diagram of the Six Spot Step Test.

**Figure 2 brainsci-11-00201-f002:**
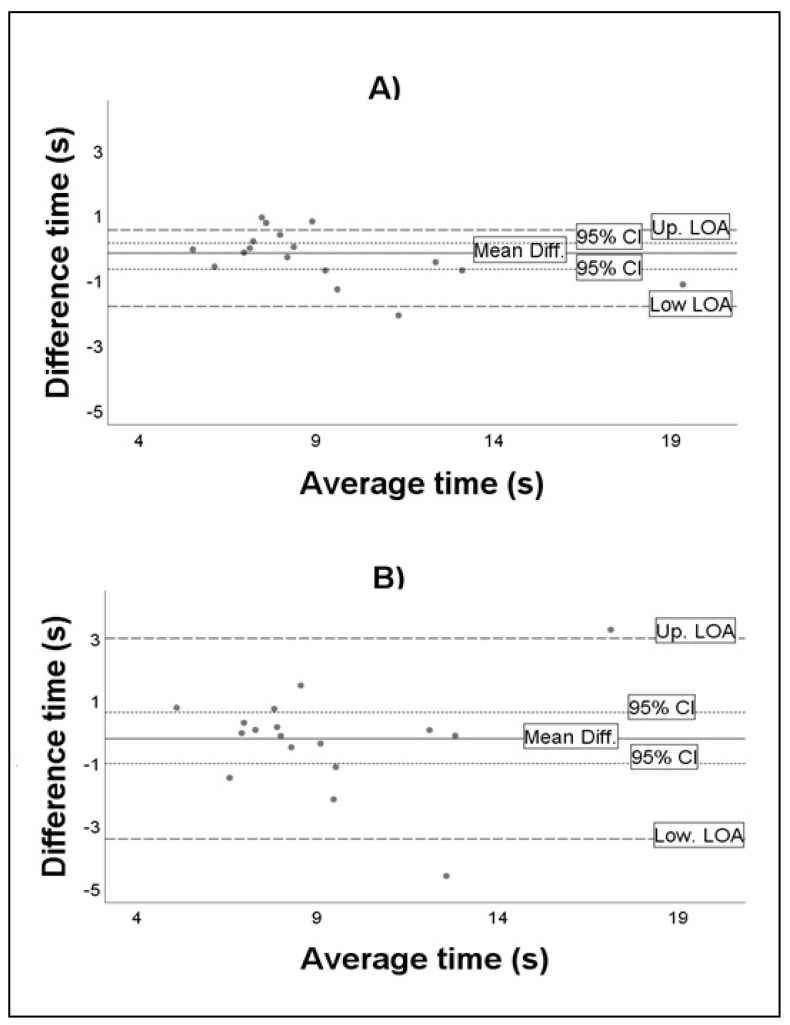
Bland–Altman plots of relative within-day agreement (**A**) and relative day-to-day agreement (**B**).

**Table 1 brainsci-11-00201-t001:** Descriptive analysis of the sample (mean ± standard deviation).

Variable	All (*n* = 18)	Men (*n* = 11)	Women (*n* = 7)
Age (years)	31.8 ± 9.9	30.5 ± 9.3	34.1 ± 10.7
Height (cm)	163.7 ± 0.1	165.6 ± 0.1	162 ± 0.1
Weight (kg)	74.1 ± 24.5	65.6 ± 14.4 *	88.4 ± 31.9 *
BMI (kg/m^2^)	27.4 ± 8	23.7 ± 3.9 *	33 ± 9.6 *
MMSE (points)	25.2 ± 6	27.9 ± 6.4	23.4 ± 5.7
TUG (s)	8.4 ± 2.4	7.7 ± 2.6	9.3 ± 2.1
FST (s)	9.3 ± 3.7	8.1 ± 2.9	11.2 ± 4.4
FRT (cm)	22.2 ± 11.5	22.8 ± 12.4	21.2 ± 11.1
LRT–R (cm)	11.8 ± 5.2	13.6 ± 4.4	8.5 ± 5.4
LRT–L (cm)	12.6 ± 5.1	14.3 ± 4.9	10 ± 4.8
BBS (points)	49.9 ± 6.4	50.8 ± 6.8	48.6 ± 6.2
ICAP–GF (points)	81.4 ± 6.8	81.1 ± 7.1	81.9 ± 6.4

BMI: body mass index; MMSE: Mini-Mental state examination; TUG: Timed up & go; FST: Four square test; FRT: Functional reach test; LRT-R: Lateral reach test–right; LRT–L: Lateral reach test–left; BBS: Berg balance scale; ICAP-GF: Inventory for Client and Agency Planning–General functionality. *t*-test between sexes: * *p* < 0.05.

**Table 2 brainsci-11-00201-t002:** Six Spot Step Test results using a stopwatch and video camera.

	All (*n* = 18)		Men (*n* = 11)		Women (*n* = 7)	
	$\bar{x}$ ± SD	95% CI	$\bar{x}$ ± SD	95% CI	$\bar{x}$ ± SD	95% CI
**Chronometer**						
First test	9.1 ± 3.2	[7.4–10.7]	8.3 ± 2.1	[6.8–9.9]	10.1 ± 4.2	[6.2–14]
Second test	9.4 ± 3.6	[7.5–11.2]	8.9 ± 2.6	[7–10.7]	10.1 ± 4.8	[5.6–14.5]
Third test	9.3 ± 3	[7.8–10.9]	9.3 ± 3	[7.2–11.5]	9.3 ± 3.3	[6.2–12.3]
**Video Camera**						
First test	8.4 ± 3.3	[6.7–10.1]	7.8 ± 2.4	[6.1–9.5]	9.2 ± 4.4	[5.2–13.3]
Second test	8.5 ± 3.2	[6.9–10.1]	8.2 ± 2.4	[6.2–10.2]	8.9 ± 3.8	[5.4–12.3]
Third test	8.8 ± 3.1	[7.1–10.4]	8.7 ± 3	[6.5–10.8]	9 ± 3.5	[5.3–12.6]

**$\bar{x}$**: mean; SD: standard deviation; 95% CI: 95% confidence interval.

**Table 3 brainsci-11-00201-t003:** Reliability test re-test for the SSST with a chronograph and video camera.

	ICC	95% CI	CV (%)	SEM	Mean Difference	95% CI	Upper LOA	Lower LOA
**Chronometer**								
Within-day agreement	0.97	0.92–0.99	4.69	0.59	−0.3	−0.4–0.11	1.26	−1.85
Day-to-day agreement	0.86	0.66–0.95	7.3	1.16	−0.26	−1.1–0.59	2.96	−3.47
**Video Camera**								
Within-day agreement	0.96	0.9–0.99	6.71	0.62	−0.99	−0.59–0.4	0.83	−2.8
Day-to-day agreement	0.88	0.71–0.96	6.79	1.07	−0.32	−1.17–0.52	2.77	−3.42

ICC: intraclass correlation; 95% CI: 95% confidence interval; CV: coefficient of variation; SEM: standard error of measurement; LOA: limit of agreement.

**Table 4 brainsci-11-00201-t004:** Similarity of measurements between subjects (data provided: r value).

Chronometer	Video Camera		
	First Test	Second Test	Third Test
First test	0.976 *	0.941 *	0.876 *
Second test	0.973 *	0.975 *	0.930 *
Third test	0.844 *	0.928 *	0.984 *

* *p* value of correlation test < 0.001.

**Table 5 brainsci-11-00201-t005:** Linear regression models for the different clinical postural control tests (continuous variables) adjusted by age.

Variables Included	Chronometer			Video		
	B	SE	*R* ^2^	B	SE	*R* ^2^
TUG	0.652	0.302	0.25 *	0.731	0.307	0.29 *
FSST	0.765	0.123	0.76 ***	0.837	0.107	0.83 ***
FRT	−0.143	0.06	0.27 *	−0.148	0.064	0.26 *
LRT–R	−0.329	0.133	0.29 *	−0.283	0.149	0.19
LRT–L	−0.068	0.16	0.01	−0.082	0.168	0.02
BBS	−0.303	0.1	0.38 **	−0.315	0.105	0.33 **

B: regression coefficient; SE: standard error; R^2^: coefficient of determination; TUG: Timed up & go; FST: Four square step test; FRT: Functional reach test; LRT-R: Lateral reach test–right; LRT–L: Lateral reach test–left; BBS: Berg balance scale. * *p* < 0.05; ** *p* < 0.01; *** *p* < 0.001.

## Data Availability

The data presented in this study are available on request from the corresponding author.

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
