# Peer review of "Reliability and Validity of the Six Spot Step Test in People with Intellectual Disability"

_brainsci, 2021, doi:10.3390/brainsci11020201_

Round 1
Reviewer 1 Report
The authors investigated reliability and validity of the Six Spot Step Test.
Introduction included relevant overview of literature, statistical analyses were conducted properly, the results are discussed adequately.
I have several minor points.
lines 61 -65 The aims and hypotheses were presented more clearly. Maybe, to present each hypothesis in a separate sentence.
I do not think that Figure 1 is necessary, because the procedure of selection of the participants is clearly explained in the text. Instead, I recommend to add a figure with schematic representations of the tasks, because procedures of the Six Spot Step Test are not known to all readers. Maybe that also a photograph would be useful.
There are several typos. There is a word „Crhonograph“ in tables 2,3, 4 and 5. It should be „chronometer“ probably.
Author Response
Dear Editor and Reviewer of Brain Sciences:
Thank you very much for your suggestions and contributions to improve the quality of the manuscript. Following your indications, we respond, point by point, to the reviewers' comments.
In the text, all the modified or added sentences have been written in red to facilitate the correction by the reviewers.
- Lines 61-65: The aims and hypotheses were presented more clearly. Maybe, to present each hypothesis in a separate sentence.
The authors have rewritten the objectives and hypotheses.
- I do not think that Figure 1 is necessary, because the procedure of selection of the participants is clearly explained in the text. Instead, I recommend to add a figure with schematic representations of the tasks, because procedures of the Six Spot Step Test are not known to all readers. Maybe that also a photograph would be useful.
Figure 1 has been replaced by another that schematically represents the Six Spot Step Test.
- There are several typos. There is a word „Crhonograph“ in tables 2,3, 4 and 5. It should be „chronometer“ probably.
The manuscript has been reviewed and corrected. Furthermore, the misuse of the term “Crhonograph” has also been removed.
Once again, thank you very much for the time spent and the interest shown in this work; as well as in the positive evaluations you have given of it.
Receive a warm greeting,
The authors.
Reviewer 2 Report
Dear authors
Thank you for submitting the manuscript entitled “Reliability and validity of the Six Spot Step Test in people with intellectual disability” for consideration for publication in Brain Sciences journal. This manuscript provides a new piece of knowledge for assessing balance in people with intellectual disabilities (PwID)
As a general comment, I don’t understand the inclusion of two methods for assessing the SSST performance (i.e. chronograph vs videorecording) since the authors recognized the time required for obtaining the test outcome with the video recording method. I suggest a better rationale about the implementation of those methods to strengthen the research interest of this study.
Please, see below my specific comments about this manuscript:
Line 17: insert “the” before “Side Spot Step”
Line 17: I suggest using the term “people”, “participants” or “persons” instead of “patients”. Your sample does not pertain to a healthcare setting.
Line 19: you should include the number of “clinical tests” used in your study.
Line 21: “the standard error of measurement” (you missed “the” and “of”)
Line 23: insert “the” before “Six Spot Step”
Line 26: remove from the keywords “Six Sport Step” and “Intellectual disability”. Both words appear in the title and you should use different words for a better indexation in databases.
Lines 41-44. I suggest a more-specific placement of the references [2, 12-14], that is, just after each of the tests/scales included in the sentence.
Lines 45-46. You describe a set of features of the tests but mixing both strengthens and weakness. I suggest explaining them with a better flow.
Lines 47-48. You are talking about the complexity of the SSST but: Is this test comprehensive for PwID? Do you have previous evidence before its implementation with your sample?
Lines 49-55. I think this paragraph would continue with the previous one.
Lines 56-60. I suggest the authors include a reference to the “dual-task” paradigm to strengthen the complexity of the SSST for PwID.
Lines 61 and 64. See my previous comment about “patients”.
Line 74. “60‒90” instead of “60-90”
Figure 1: Insert a blank space before “(n=34)”
Line 90. I suggest including the description of the test early in the Procedures subsection or having a specific subsection before Procedures. If you know the tests, you can better understand their implementation.
Lines 106-108. It was mandatory kicking the blocks with the feet area that you are describing. I think this is an issue that may influence in PwID.
Lines 114-115. How many trials of attempts they have in case of failures?
Lines 117-118. Did you apply all the tests in the same order?
Lines 122-123. There are also reliability studies with people with Down Syndrome like as https://pubmed.ncbi.nlm.nih.gov/31357594/
Line 130. “0.73‒0.98” instead of “0.73-0.98”
Lines 131 and 137. Abbreviations for FRT and LRT should be described, respectively.
Lines 131-145. I am wondering about the inclusion of the FRT and LRT test in the battery since thy are tests that are conducted in a sitting position. Why do you expect relationships with your standing test and your SSST as a gold standard?
Line 150. With which population there is good reliability for the BBS?
Line 154. “deviations” instead of “deviation”
Lines 156-157. The outcome of the videorecording is not clear. Please, clarify.
Line 173. How did you control the age in the regression model? Is age a covariable?
Lines 184-188. Please, review the line spacing of the Table 1 footnote.
Tables 2, 3 and 4: “Chronograph” instead of “Crhonograph”
Line 202. Why did not use SEM% for a better comparison with the reported CV scores?
Lines 207-208. Remove (A) and (B); which is the linkage?
Table 3: “Within-day” instead of “Within day” (x2)
Table 3: “Upper” and “Lower” instead of “Up” and “Low”
Table 3: Remove the first line of the table: i.e. Relative / Absolute
Line 247. “observations” instead of “observation”
Table 4. “0.930*” instead of “0.93*”
Lines 252-253: r values should include two decimals.
Lines 256-257. I suggest including a statement about the percentage of explained variance of your regression model.
Lines 272-275. Please, review the line spacing of Table 5 footnote.
Line 278. Same comment I said before about using “patient”
Lines 291-292. I think your protocol (or future studies” would be improved using a time gate for establishing a more precise beginning of the start/finish moments of the test.
Line 306. Remove “…” if you include “etc.” in the sentence.
Line 315. “postural control” appears twice. Would you change by “balance” at the end of the sentence?
Line 322. Same comment I said before about using “patient”
Lines 320-324. This sentence is too long. I suggest a division in two shorter sentences.
Lines 329-332. As I said before, in the Introduction, there is a no clear rationale about the inclusion of postural control tests that are performed in a sitting position. A better rationale of the tests included in your battery should be included.
Lines 333-334. Although you state as a limitation of your study the sample size: Did you powered the sample size? I suggest including a calculation of the statistical power (e.g. using GPower or similar tools)
References 24 and 26. Which are the end pages for these studies?
Author Response
Dear Editor and Reviewer of Brain Sciences:
Thank you very much for your suggestions and contributions to improve the quality of the manuscript. Following your indications, we respond, point by point, to the reviewers' comments.
In the text, all the modified or added sentences have been written in red to facilitate the correction by the reviewers.
- As a general comment, I don’t understand the inclusion of two methods for assessing the SSST performance (i.e. chronograph vs videorecording) since the authors recognized the time required for obtaining the test outcome with the video recording method. I suggest a better rationale about the implementation of those methods to strengthen the research interest of this study.
The authors have expanded the Introduction justifying the interest of including both measurement methods.
- Line 17: insert “the” before “Side Spot Step”.
I suggest using the term “people”, “participants” or “persons” instead of “patients”. Your sample does not pertain to a healthcare setting.
Line 19: you should include the number of “clinical tests” used in your study.
Line 21: “the standard error of measurement” (you missed “the” and “of”).
Line 23: insert “the” before “Six Spot Step”.
The authors have done all these corrections belonging to the Abstract.
- Line 26: remove from the keywords “Six Sport Step” and “Intellectual disability”. Both words appear in the title and you should use different words for a better indexation in databases.
The authors have eliminated both key words and have included: Motor control; Balance; Evaluation.
- Lines 41-44. I suggest a more-specific placement of the references [2, 12-14], that is, just after each of the tests/scales included in the sentence.
The authors have replaced the citations next to the test to which they refer.
- Lines 45-46. You describe a set of features of the tests but mixing both strengthens and weakness. I suggest explaining them with a better flow.
The authors have rewritten that sentence.
- Lines 47-48. You are talking about the complexity of the SSST but: Is this test comprehensive for PwID? Do you have previous evidence before its implementation with your sample?
There are no scientific publications about it. This is the first time the SSST has been applied to people with intellectual disabilities.
- Lines 49-55. I think this paragraph would continue with the previous one.
Lines 56-60. I suggest the authors include a reference to the “dual-task” paradigm to strengthen the complexity of the SSST for PwID.
Lines 61 and 64. See my previous comment about “patients”.
Line 74. “60‒90” instead of “60-90”.
The authors have corrected all these errors.
- Figure 1: Insert a blank space before “(n=34)”.
Figure 1 has been replaced by a different one (by the demand of Reviewer No. 1).
- Line 90. I suggest including the description of the test early in the Procedures subsection or having a specific subsection before Procedures. If you know the tests, you can better understand their implementation.
Lines 106-108. It was mandatory kicking the blocks with the feet area that you are describing. I think this is an issue that may influence in PwID.
Lines 114-115. How many trials of attempts they have in case of failures?
Lines 117-118. Did you apply all the tests in the same order?
All these details have been corrected or added to the manuscript.
- Lines 122-123. There are also reliability studies with people with Down Syndrome like as https://pubmed.ncbi.nlm.nih.gov/31357594.
The indicated reference has been added.
- Line 130. “0.73‒0.98” instead of “0.73-0.98”.
That detail has been corrected.
- Lines 131 and 137. Abbreviations for FRT and LRT should be described, respectively.
These abbreviations are defined the first time those terms are used in the manuscript (in the Introduction section).
- Lines 131-145. I am wondering about the inclusion of the FRT and LRT test in the battery since thy are tests that are conducted in a sitting position. Why do you expect relationships with your standing test and your SSST as a gold standard?
These tests are not used as a gold standard. The authors decided to include them because they are two of the most frequently used tests for the evaluation of balance and postural control of the trunk in people with intellectual disabilities. This aspect is better explained in the new version of the Introduction to the manuscript.
- Line 150. With which population there is good reliability for the BBS?
Line 154. “deviations” instead of “deviation”.
Lines 156-157. The outcome of the videorecording is not clear. Please, clarify.
All these details have been corrected or added to the manuscript.
- Line 173. How did you control the age in the regression model? Is age a covariable?
The models were adjusted according to the age variable. This detail has been explicitly specified in the manuscript.
- Lines 184-188. Please, review the line spacing of the Table 1 footnote.
Tables 2, 3 and 4: “Chronograph” instead of “Crhonograph”.
Lines 207-208. Remove (A) and (B); which is the linkage?
Table 3: “Within-day” instead of “Within day” (x2).
Table 3: “Upper” and “Lower” instead of “Up” and “Low”.
Table 3: Remove the first line of the table: i.e. Relative / Absolute.
Line 247. “observations” instead of “observation”.
Table 4. “0.930*” instead of “0.93*”.
Lines 252-253: r values should include two decimals.
Lines 272-275. Please, review the line spacing of Table 5 footnote.
Line 278. Same comment I said before about using “patient”.
Lines 291-292. I think your protocol (or “future studies” would be improved using a time gate for establishing a more precise beginning of the start/finish moments of the test.
Line 306. Remove “…” if you include “etc.” in the sentence.
Line 315. “postural control” appears twice. Would you change by “balance” at the end of the sentence?
Line 322. Same comment I said before about using “patient”.
Lines 320-324. This sentence is too long. I suggest a division in two shorter sentences.
Lines 329-332. As I said before, in the Introduction, there is a no clear rationale about the inclusion of postural control tests that are performed in a sitting position. A better rationale of the tests included in your battery should be included.
Lines 333-334. Although you state as a limitation of your study the sample size: Did you powered the sample size? I suggest including a calculation of the statistical power (e.g. using GPower or similar tools).
References 24 and 26. Which are the end pages for these studies?
All these details have been corrected or added to the manuscript.
Once again, thank you very much for the time spent and the interest shown in this work; as well as in the positive evaluations you have given of it.
Receive a warm greeting,
The authors.
Round 2
Reviewer 2 Report
Dear Authors
Thank you for including all the suggestions sent in my previous revision.
Please, review in Line 179 "eldely", since it should be "elderly".
Kind regards,